# Multidimensional Measures and the Extra Costs of Disability: How Are They Related?

**DOI:** 10.3390/ijerph20032729

**Published:** 2023-02-03

**Authors:** Mónica Pinilla-Roncancio

**Affiliations:** School of Medicine, Universidad de los Andes, Bogotá 110111, Colombia; mv.pinilla@uniandes.edu.co

**Keywords:** disability, extra cost, poverty, deprivation, multidimensional measures

## Abstract

People with disabilities are more likely than individuals without disabilities to face higher levels of deprivation and multidimensional poverty, and those deprivations might be associated with the extra costs of living with a disability. However, there has not been an analysis of how multidimensional poverty measures are related to the extra costs of disability or whether these measures can be used as a proxy of the standard of living in the analysis of the extra costs of disability. This paper aims to analyse whether multidimensional poverty measures can be used to study the extra costs of disability and, based on the capability approach, how multidimensional poverty is related to the extra costs of disability. This paper discusses theoretical, technical, and methodological aspects to be considered when studying the relationship between extra costs and multidimensional poverty, and we used data from Chile and Nigeria to illustrate this relationship. We conclude that when analysing the extra costs of disability, multidimensional measures might be an option; however, it is necessary to clearly stablish the relationship among income, deprivation, and the extra costs of disability.

## 1. Introduction

The design of inclusive social protection programmes for people with disabilities requires information to be obtained from a variety of sources, and it necessitates, for example, disaggregating data by disability status, understanding the socioeconomic characteristics of persons with disabilities, and recognising their specific needs. The analysis of the disability-related costs faced by individuals and households has become a priority for a range of countries in order to define the scope and level of support that people with disabilities should receive and to estimate their levels of poverty.

Studies analysing the extra costs of disability have used a variety of methodological approaches to study the relationship among extra cost, disability, and income [1,2]. One approach is to ask people directly about their spending, both actual and needed. In this approach, individuals with disabilities are asked to provide information about their expenditures and to identify the goods and services that they need but might not be able to afford. 

Good and Services Required (GSR) uses focus groups and qualitative methods to provide important information on the main extra costs of disability and how different types of functional limitations are associated with different costs and coping strategies. However, only a few countries have implemented this method and use it to inform policy decisions related to disability benefits. GSR uses expert groups of people with disabilities, who provide a comprehensive list of goods and services that they need and require [3]. 

The Standard of Living (SoL) method estimates the extra direct costs of disability and calculates the extra income that a household or an individual with disabilities needs in order to have the same standard of living as that of a person without disabilities [4]. A systematised review found that eight articles have been published using this method [1], and a new review, which updated this result, found 18 new articles using the SoL that were published after 2015 [2]. 

When applying the SoL method, two main variables have been used to define the standard of living. The first follows Filmer et al. and uses principal components to compute a wealth index, which is a proxy of consumption [5]. The second type of variable is a subjective measure of material deprivations, which asks questions related to material deprivations. Those questions, commonly used by researchers in Europe, follow a material deprivation approach, which assumes that deprivations in different aspects of life are related mainly to a lack of income or monetary resources [6]. 

Both variables impose assumptions on the relationship between income and disability. On the one hand, when the wealth index is used, it is assumed that a reduction in income will be directly related to a reduction in the value of the wealth index. On the other hand, when using subjective questions about material deprivations, we assume that the main reason for not having an item is lack of money. Although these assumptions have theoretical and empirical justifications, it is important to consider that in the case of disability, the relationship between extra cost and income is mediated by an individual’s opportunities to generate and use income. 

In the past decade, there has been an increase in the number of studies analysing the levels of multidimensional poverty experienced by people with disabilities and their families. Most studies have found that they are more likely to be poorer and to face a higher number of deprivations, compared with persons without disabilities and their families [7,8,9,10,11,12,13,14]. Multidimensional measures designed at the individual level, including indicators that might be affected by a disability, reveal that people with disabilities are usually disadvantaged compared with people unaffected by disability and other household members without disabilities. 

In the context of the analysis of extra costs, multidimensional poverty measures can offer an opportunity to capture how disability affects other aspects of life and how those might increase a person’s risks of facing extra costs and becoming poor. Until now, multidimensional poverty measures have not been used to analyse the extra costs of disability and have not been evaluated to see if these measures are a good proxy of the standard of living and how the analysis can be implemented. In addition, it is important to define how the extra costs of disability increase the risk of deprivation and how being deprived in a specific indicator increases the risk of facing higher extra costs associated with disability and an increased risk of deprivation in other indicators. This is how the disability–deprivation/poverty cycle starts and persists. 

This paper has two main purposes: first, to analyse whether multidimensional measures can be an option in the study of the extra costs of disability using the SoL method and second, to discuss how the additional costs related to disability are associated with multidimensional poverty. The article first introduces the SoL method and the most important theoretical and methodological assumptions behind it. It then considers what multidimensional poverty measures are and how they are designed and computed. This is followed by a discussion about the relationship between multidimensional poverty and the extra costs of disability; we ask which theoretical and methodological aspects need to be considered when thinking of using a multidimensional poverty measure in the analysis of the extra costs of disability, and we offer an illustration of how these aspects should be considered in the analysis. 

## 2. Standard of Living Method

The Standard of Living (SoL) method aims to analyse the relationship between living with a disability and household consumption levels [4]. The method assumes that an income transfer can compensate for the reduction in a household’s standard of living and that the existence of a disability causes those reductions. It also assumes that households with members with disabilities have different conversion factors. Therefore, how income is converted to a specific standard of living depends on the type and the severity of the disability. This method assumes that the extra costs of disability are the additional income required to maintain the same standard of living as a household without a member living with a disability [4]. 

One of the most important assumptions of this model is that two individuals with the same income level experience different living standards if one has a disability and the other does not. This is assumed to be the case because the person with disabilities will spend more on the consumption of disability-related items and will therefore reduce the overall consumption of other items, an aspect that will affect his/her standard of living [4]). 

The concept of standard of living is fundamental for this method and its operationalisation. Therefore, the variable that captures this concept should reflect objects of value [15] and it should be influenced by the economic means of individuals or households. Thus, the variable of the standard of living should fulfil two main desired characteristics: (1) capture the consumption of items and assets and (2) be influenced by the individual’s or household’s level of income. It is important to note that the concept of standard of living is different from the one of well-being or poverty. The concept of well-being incorporates aspects which, in some cases, cannot be measured or compensated for by income and may be subjective by nature [15]. In the case of poverty, the purpose is to capture the minimum level of economic resources or capabilities that a person needs to exhibit to be considered poor. Here, the analysis focuses on those who are deprived and, for different reasons, do not have access to services and opportunities [16,17]. 

The SoL method only aims to capture the average direct costs of disability. It assumes that people with disabilities have different needs from those unaffected by disability and that therefore they prioritise those needs and reduce their consumption of nondisability-related items. However, the relationship between income and consumption is not as direct as the SoL method assumes. Indeed, although income is one of the most important determinants of consumption, credit accessibility, psychological factors, and the ability to pay for and buy products are also significant determinants of consumption [18]. 

In conclusion, the SoL method allows for the analysis of the relationship among standard of living, income, and disability. However, it ignores important determinants of consumption, such as individual needs, preferences, access to credit or savings, and the supply of services that enable people to consume different goods. As a result of ignoring these important aspects, people with disabilities living in low-income countries have presented lower extra costs of disabilities compared with disability-affected people living in countries with higher incomes and levels of human development. However, these results do not reflect the reality that confronts people living with disabilities, and given the limitations of the SoL, it is not possible to analyse the potential effect that extra costs of disabilities have on the lives of individuals with disabilities and their families. Therefore, a measure that is sensitive to the context and can be tailored to it is necessary in order to capture the real effect of the extra costs of disability on the standard of living of individuals.

## 3. Multidimensional Poverty Measures

A range of perspectives for understanding poverty has been adopted in the literature. According to Ringen (1988), measures of poverty can be divided into direct and indirect measures [19]. The direct type is related to aspects of well-being affected by poverty, and it captures the lack of access to services and opportunities. Indirect measures include income and consumption. 

Multidimensional poverty measures are considered to be direct measures of poverty because they capture the actual deprivations that a person or a household faces in a specific moment. Multiple methods to measure multidimensional poverty exist in the literature. However, in the past decade, the Alkire–Foster (AF) method has become one of the most popular and is frequently used to measure multidimensional poverty [20]. The AF method is based on a counting approach, which defines a deprivation profile for each individual (or household). This method follows the two steps suggested by Sen (1979) to compute a poverty measure. First, it identifies who the poor are, and then it creates an aggregated measure. In addition, the AF method uses a double cut-off approach, which identifies individuals deprived in each indicator and then individuals who are multidimensionally poor [21]. 

The AF method computes three main measures: the incidence of multidimensional poverty (H), its intensity (A), and the adjusted headcount ratio (M_0_). The incidence represents the percentage of people who are multidimensionally poor; the intensity represents their average number of deprivations, and M_0_ represents the percentage of potential deprivations that poor individuals face, allowing for the total possible number of deprivations of the society in question [20]. 

Although disability is an individual phenomenon, in some cases, multidimensional poverty has been computed at the household level, and then measures have been disaggregated between households with and without members with disabilities [9,11]. Other studies have computed measures at the individual level [7,8,10,12,13,14], and others have been designed specifically for persons with disabilities [22]. Multidimensional poverty measures, or disability-specific measures, have included indicators related to education, health, employment, and, in some cases, discrimination or social and/or attitudinal barriers. Given that these measures use the individual as the unit of identification, it is possible to analyse intrahousehold inequalities and differences relating to age, sex, and other individual characteristics.

## 4. Multidimensional Poverty and Its Relationship with the Extra Costs of Disability

To date, no study has analysed the relation between the extra costs of disability and the multidimensional poverty levels of people with disabilities. However, there is evidence that such people face higher barriers to accessing education [23,24], health care services [25,26], and employment [27] and that those barriers might be associated with the cost of transportation [28] and the need for support and assistive devices [29], among other factors. In this section, we will explore the potential causal links among four sources of extra costs of disabilities and multidimensional poverty. First, we will analyse how transportation costs might be associated with higher levels of deprivation in different indicators. Second, we will analyse how assistive-device costs can increase health and education deprivations. This is followed by an analysis of how additional health care costs and accessibility costs might increase the probability of deprivation as measured by a range of indicators relating to health, education, employment, and living standards. 

### 4.1. Transportation Costs 

Education: Higher costs of transportation can reduce the probability of access to education services [30,31,32]. Therefore, a person can face higher deprivations in terms of years of schooling and school attendance; furthermore, high transportation costs increase the probability that children who are currently attending school are lagging behind their classmates or placed in a grade lower than the one expected for their age (school lag) because they might have missed several school days, and they have a higher likelihood of dropping out of school and/or having to repeat school years. 

Health: Higher transportation costs can increase the probability of accessing health care services when needed [26,33]. This will increase the risks of unsatisfied health care needs, the deprivation of access to health care, and the risks of living with chronic diseases and not having access to medicines or adequate treatments [26,34,35].

Employment: Higher transportation costs can increase the probability of persons with disabilities being unemployed or outside the labour market. In addition, they might be associated with informal employment, given that persons with disabilities would have more flexibility and can work close to their homes [36].

### 4.2. Human Support and Assistive-Device Costs 

Education: If individuals need to invest in human support or assistive devices, they might have a lower probability of attending school, because they do not have the resources to buy assistive devices, and the lack of such devices will become a barrier to attending school. Therefore, their levels of educational attainment will be lower. As with transportation costs, individuals with disabilities will have higher chances of dropping out of school or having to repeat school years. 

Health: Facing unaffordable assistive-device costs can increase the probability that a person does not have access to health care services and will therefore experience lower health care outcomes, with a higher probability of being exposed to chronic diseases and not receiving the correct treatment [26]. 

### 4.3. Health Care Costs 

Health: In the case of health care, people with disabilities have higher levels of need [26,37], which may be unsatisfied. For example, they will have lower access to medicines and preventative services, such as vaccination, prenatal care, and sexual health and reproductive care [26,38,39]. 

### 4.4. Accessibility Costs 

Education: Higher accessibility costs might increase the risk of children with disabilities not attending school, thus increasing deprivation in this indicator. In addition, this will have a negative effect on the school attainment of adults and a potential effect on school lag for school-aged children. 

Health: Higher accessibility costs might reduce the chances of persons with disabilities accessing health care services, such as nutrition, vaccination, and prenatal and postnatal care, and they will have a higher risk of being deprived in access to medicines or medical treatments. 

Employment: Higher accessibility costs might reduce the probability of persons with disabilities finding work and having a good-quality job. Therefore, they are more likely to be deprived in respect of indicators, such as unemployment, informal employment, underemployment, and NEET (Young People Not in Education or Training). 

Living standards: In this dimension, it is important to consider access to the services and their use. Higher accessibility costs will increase the risk of deprivation in access to a clean water source and improved sanitation, as well as to the Internet and other modern technology.

A multidimensional poverty measure can include deprivations in aspects that may be affected by the extra costs of disability, for example, considering a measure with four dimensions and 11 indicators (Figure 1). Therefore, a person with disabilities facing higher levels of extra costs probably would have higher levels of deprivation and thus be considered multidimensionally poor. Nevertheless, depending on the context and the types of opportunities and services available, people with disabilities will have higher or lower costs associated with the achievement of different indicators. For example, in a country with limited access to education (caused, for example, by a limited number of schools), a percentage of children with and without disabilities will not be able to access education, and therefore the costs associated with accessing education for children with disabilities and their families will not be as high as expected, not because they do not have to meet extra costs, but because there is a general lack of access to education. This last point can be captured by multidimensional poverty measures. 

It is not possible to assume that higher levels of deprivation are only the result of the extra costs that a person might face. Indeed, deprivations in access to school attainment or attendance can result from other factors, such as attitudinal and physical barriers, which go beyond a person’s (or a household’s) ability or opportunity to pay for different services. 

In addition, although the evidence suggests that on average individuals with disabilities face higher deprivations in one or more indicators and usually their levels of deprivation are more severe than those of people without disabilities, this will not be the case for all people with disabilities in a specific context. Finally, using a multidimensional poverty measure as a proxy of standard of living would not make it clear how income compensation can reduce deprivation, especially in cases where the deprivation is created by social or environmental factors, which go beyond an individual’s own control. For example, even if households that include disability-affected members are compensated with a certain amount of income, that will not compensate them for the general lack of accessible transportation systems or the lack of schools or health providers in the community.

### 4.5. Aspects to Consider When Using a Multidimensional Measure in the Analysis of the Extra Costs of Disability

A range of aspects should be discussed when considering whether a multidimensional poverty measure can be used as a proxy for the standard of living in the analysis of the extra costs of disability. First, theoretical arguments are associated with how the concept of standard of living is understood. The second aspect is related to understanding the relationship between income and deprivation, the role of income in reducing deprivation, and how this role can be (or not be) captured in the analysis of the extra costs of disability, using a multidimensional measure. The third and final aspect is to understand what a multidimensional poverty measure represents, what aspects of poverty are captured, and the main aspects to consider when designing a measure for people with disabilities. 

### 4.6. Theoretical Arguments

The first aspect to consider is how poverty and standard of living are defined. Under the capability approach, which understands well-being in terms of capabilities and functionings, where capabilities are the doings and beings that people can achieve if they so choose and functionings are capabilities that have been realised ‘poverty’ is the lack of basic capabilities [17], and ‘standard of living’ is the ability to achieve various personal conditions [15]. In this context, although poverty can be associated with a lower standard of living, this association is not direct and will depend on how poverty is defined and on the list of basic capabilities included in the analysis. In addition, when thinking about multidimensional poverty, it is important to understand that a person who is multidimensionally poor suffers a range of deprivations simultaneously; however, the aspects in which the person is deprived depend on his/her personal situation and how those deprivations interact with the additional opportunities open to the person in question. 

The SoL method assumes that a person with a disability has a lower standard of living because she/he reduces her/his consumption of different goods because of the need to afford disability-related expenditures. Thus, we are assuming that there is a direct relationship between disability and consumption and that individuals with and without disabilities have the same preferences and will decide to consume similar items. However, when this assumption is analysed in terms of the concept of multidimensional poverty, an important role is played by aspects related to the opportunities available in the society, how services are provided, and the preferences of individuals and families. For example, it is difficult to assume that an individual will not have access to clean water because he/she needs to increase his/her disability-related consumption. In this case, access to a clean water source will be mediated by the existence of piped water services in the neighbourhood or the provision of other protected water sources, such as a protected spring or well, and although there is a possibility that people with disabilities do not have access to the source of water, this barrier is not the result of favouring the consumption of water above the consumption of other disability-related goods. 

### 4.7. Income and Deprivation: Are They Directly Related?

Although there is a relationship between income and deprivation, being deprived in nonmonetary aspects is mediated by other factors, which might reduce or increase the potential effect of lack of income. In terms of the capability approach, income is seen as a means but not as an end. In addition, it is recognised that in order to be able to transform income into achievements, conversion factors play a major role. Conversion factors—which influence how people can turn means into ends—can be classified as personal, social, and environmental. The first one depends on the person’s circumstances, for example, disability. The second factor is social conversion aspects related to the society where the person lives; one example is class or caste. The third and final group consists of environmental conversion factors that capture the physical environment where the person lives, such as geographical location [40]. 

Policy makers can change conversion factors, mainly social and environmental factors, by implementing social policies to reduce environmental barriers or discrimination, which might increase the possibilities of social participation. Moreover, policies can also target people with different individual conversion factors, such as people with disabilities, and provide access to services and benefits to improve their quality of life. 

Given the existence of conversion factors, income compensation is not the only or best solution to reduce deprivations. Indeed, an income transfer cannot be effective in the case of countries that do not provide inclusive services, such as education or health care. Amartya Sen gives an example of a conversion factor directly related to disability [41,42]. Under the capability approach, people with disabilities face a double handicap: one relating to the lack of income that they and their families receive (income handicap) and a conversion handicap relating to the difficulties confronting people with disabilities who must convert their available income into essential goods and services. Therefore, people with disabilities can have the same level of income as people unaffected by disability or a level of income that produces a similar standard of living, but without the existence of social and environmental transformations, it will not be possible to obtain the same levels of satisfaction. 

In the context of multidimensional poverty, the role of environmental and social factors is vital, because two individuals with the same levels of multidimensional poverty may have a different combination of deprivations and, most importantly, the circumstances that created those deprivations might be different. Therefore, it will be unfair to say that the same income compensation can compensate for deprivations in health (for example, access to health care) or in education (for example, years of schooling).

In conclusion, the magnitude of the potential effect of income on deprivation depends on a range of factors that might affect how income affects deprivations. Individual, social, and environmental factors play an important role in defining whether a person is deprived and in which particular aspects of development. Thus, it is fundamental to incorporate those aspects into the analysis in order to capture the problem’s complexity and potential circumstances. 

### 4.8. Multidimensional Poverty Measures: What Do They Represent?

As discussed above, multidimensional measures are direct measures of poverty, which include aspects related to deprivations in various aspects of life. Most measures include deprivations in terms of health care, education, living standards, and employment. The number and type of indicators depend on the purpose of the measure and the available data. One limitation of the multidimensional measures is that no measure can include all dimensions of poverty, and there are trade-offs between indicators. 

Additionally, depending on the weight structure, indicators considered in multidimensional poverty measures can be defined as supplements or complements. For example, consider the global multidimensional poverty index, which has three dimensions and ten indicators. Each dimension has the same weight (1/3), and each indicator inside the dimension has the same relative weight. Therefore, indicators in the health and education dimensions have the same weight, and a weight almost three times than one indicator in the living standards dimension. In this context, a person who is deprived in terms of access to water, sanitation, and electricity has the same level of deprivation as a person who is deprived in terms of child mortality or school attendance. Therefore, when analysing a multidimensional poverty measure, it is important to consider what represents a specific level of deprivation. For example, if two people have a counting vector that is the number of weighted deprivations that a person faces, equal to 40%, each of them might have a different combination of indicators distributed among dimensions. Consequently, policy makers should be able to identify trade-offs among dimensions and indicators and how those are related to the extra costs of disability. In addition, it is important to identify how deprivations limit the achievement of different capabilities and functionings, how extra costs can create different deprivations, or how deprivations in one aspect create extra costs that increase the probability of being deprived in other indicators. 

## 5. Empirical Applications

Aiming to illustrate the previous discussion, we used data from Chile and Nigeria. We selected Chile because it is a high-income country, with levels of multidimensional poverty lower than 20% and with national policies to guarantee the rights of people living with disabilities. Nigeria was included in the analysis because, contrary to Chile, the country has high levels of deprivation and multidimensional poverty, and although there are policies to protect the rights of persons with disabilities, given the high levels of poverty of the country, those might have a reduced impact on the lives of persons affected by disabilities. 

In Chile, we used the Socioeconomic Conditions Survey 2017 and in Nigeria the Living Standard Survey 2019/2020. In the case of Chile, we computed the National MPI for Chile [43], and in Nigeria we computed an MPI, which followed closely the methodological decisions of the global MPI [44]. However, because of data availability, it was only possible to use eight of the ten global MPI indicators. Therefore, we included food security and health service access in the dimension of health 

Once we had computed the MPI, we estimated the extra costs of disability using the SoL method for persons 18 or older. The dependent variable was the counting vector (or the sum of weighted deprivations per household) of the multidimensional poverty measure; as independent variables, we used the following factors: the existence of a member with disability in the household (following the Washington Group (WG) definition), household income, size of the household, region, area of residence, and gender of the head of the household. In each of the countries, the model used robust standard errors. 

The results of the analysis reveal that there Is a negative relationship between increasing the number of deprivations and the income level of the household. In addition, there is a positive association between households with at least one member with disabilities and the number of deprivations that the household faces. These signs are different from the ones usually identified in the SoL method, where disability is usually associated with a lower standard of living, and the standard of living variables are positively associated with income. However, when we computed the coefficient related to the extra costs of disability, the ratio is negative, although the negative sign is produced by the negative association between income and the sum of deprivations. 

In addition, it is important to analyse the relationship between the different indicators of the MPI by levels of income and to disaggregate the MPI by disability status of the household. Table 1 presents the main results of the disaggregated analysis. In both countries (Chile and Nigeria), people with disabilities present higher levels of multidimensional poverty and deprivation in all the indicators included in the measures. 

When we disaggregated the types of deprivation faced by households with disabled members according to their income level, we found important differences between income quintiles in the case of Chile. For example, in the lowest income quintiles, households with and without members with disabilities face similar deprivations as households without members with disabilities (except in the pension and environment indicators) (Figure 2). By contrast, when we analysed and compared the percentages of households with and without members with disabilities in the highest quintile, the differences are larger compared with those in the lowest quintile (Figure 3). In the case of Nigeria, we found that there are no differences between the types of deprivation that individuals face in terms of income quintiles (Figure 4 and Figure 5). This may be related to the fact that deprivations in Nigeria are more acute and more broadly affect households with and without members with disabilities.

In both countries, the multidimensional poverty measures were computed at the household level, and they included deprivations that affected members with and without disabilities in the household. It is not possible to easily identify whether the person with disability is the one creating the deprivation. Therefore, it is not possible to assume that the higher levels of multidimensional poverty are directly related to disability and that an increase in the levels of household income would compensate for the extra costs of disability and reduce multidimensional poverty. For example, in the case of Chile, when we computed the ratio between the disability and the income coefficients, the results reveal that it is necessary to have a 35.1% income compensation because of the extra costs associated with disability. However, if we analyse the combination of deprivations faced by households with members with disabilities, we find that the two pairs of indicators with the higher percentage of households with disabilities deprived are (1) contribution to social security and years of schooling and (2) habitat and years of schooling. In both cases, it is not possible to define directly whether these deprivations are the results of a disability or were produced before the disability occurred. 

In the case of Nigeria, the ratio between the coefficients of disability and income was equal to 13.1%. Thus, households with at least one member with disabilities will need an increase in their income equal to 13.1%. In addition, in Nigeria, the types of deprivation affecting the largest percentage of persons with disabilities were (1) food security and cooking fuel and (2) cooking fuel and housing. In these four deprivations, it is not possible to assume that the deprivations are the result of the extra costs of disability and that income compensation will cover the extra costs of disability and directly reduce levels of deprivation. 

## 6. Potential Uses of Multidimensional Measures in the Analysis of Extra Costs of Disability

A multidimensional measure will enable a clearer analysis of the potential loss of welfare faced by individuals because of their disabilities. In this case, the assumption will not be whether individuals with disabilities reduce their consumption of different items because of the increase in the consumption of disability-related items; instead, the main assumption is that different factors associated with barriers that persons with disability face might reduce their well-being or increase their levels of poverty and deprivation, for example, discrimination, stigma, social exclusion, and lack of participation. 

This analysis will allow us to go beyond income compensation and understand that the extra costs of disability have a negative impact on the levels of capabilities and functioning that a person can achieve or limit the opportunities open to people living with disabilities of living the life that they would like to live. Whether realised or not, extra costs affect the opportunities of individuals with disabilities in general to access services that will facilitate income generation and their equal participation in society. The AF method provides an opportunity to design a measure that is tailored to the context. It includes indicators that capture the desirable levels of achievement in different aspects of human, social, and economic development. However, to properly capture well-being, the following procedures are necessary: Define the measure at the individual level. Thus, use the individual as the unit of identification and analysis. This will directly identify individual deprivations and achievements and allow us to assume that higher levels of deprivations or lower levels of achievements are directly related to disability or the extra costs of disability.Include indicators that reflect achievements for the whole society, disregarding individual characteristics, such as race, ethnicity, or disability, for example, the level of education that a person is expected to achieve in a specific country.Clearly define the trade-offs among indicators. For example, suppose that the measure includes five dimensions and 20 indicators equally weighted. In that case, it is important to define which indicators have a higher or lower relative importance and how those levels might affect the results of the measure.In this case, it is not necessary to define an optimal level of well-being or poverty (or to compute the incidence of well-being). Ideally, one should analyse the counting vector or the weighted sum of indicators.The challenge is then to establish how income compensation can improve lower levels of achievement in important aspects of social, human, and economic development and whether income is the only aspect that social policies can change to guarantee that persons with disabilities can participate in society. In this case, it will be important to analyse that the relationship between multidimensional measures and the extra costs of disability depends on the opportunities available in the society in question and that those opportunities will mediate in the achievement of different capabilities and functionings.

In addition, the following aspects might be considered, if this multidimensional measure is used in methods, such as SoL: It is recommended to estimate regressions by income quintiles; thus, the sample can be stratified, and an income coefficient is computed for each quintile. It is expected that in the highest quintile, the income compensation will be lower. However, because the level of individual well-being is the result not only of consuming specific assets or goods but also of the person’s access to opportunities, people with disabilities will face barriers in all income quintiles, and so they will still face lower levels of well-being.The inclusion of the education dimension should be considered, depending on the context. Endogeneity problems exist, given that higher educational levels are associated with higher income. However, it will be advisable to explore what is the contribution of this dimension to the well-being measure and to conduct robustness tests to analyse how changes in the structure affect the results.The dimension of employment aims to capture access to good-quality jobs and life–work balance, which is not directly associated with income levels. It is recommended that this dimension includes indicators related to satisfaction with employment or underemployment (inadequate jobs for their training). The objective of this dimension is to identify if the person has a job that matches his/her qualifications.Other options of indicators are social and family support, social capital, trust, and inclusion in decision making.Depending on the country’s development levels, the dimension of living standards can be included in the multidimensional measure.

Finally, it is expected that people with disabilities face lower levels of well-being because they face different social, economic, attitudinal, and physical barriers. Although it is expected that an absolute well-being measure, which uses objective indicators, will better capture the potential impacts of the extra costs of disability, it will be necessary to empirically test the measure and the relationship with the extra costs of disability.

### When to Use Multidimensional Measures to Analyse the Extra Costs of Disability?

As has been discussed, multidimensional measures can provide information related to the deprivations or achievements faced by individuals with disability and how those can vary depending on the disability status and the severity of the disability. In addition, multidimensional measures provide information on how external barriers can affect individual achievement levels. Contrary to other measures usually used to analyse the extra costs of disability (such as the wealth index), multidimensional measures are absolute measures whose main structure and relative contribution are defined on the basis of normative arguments. In the case of the wealth index, it is not possible to know which indicators or factors contribute the most to the index (unless this information is analysed). Therefore, it is not possible to know which indicators are negatively associated with disability. In addition, when using the wealth index in the SoL method, the results will reveal how much more income a household with a disability needs in order to have the same economic resources to acquire non-disability goods and services as enjoyed by a household without a disability. It implies that giving them money equivalent to the extra costs will allow them to purchase the same assets and have the same level of economic well-being. In the case of multidimensional measures, the analysis of the extra costs will tell us which potential income compensation is needed to equalise the well-being of a person with a disability. Nevertheless, this income compensation will need to be complemented by providing opportunities to access services. However, a person will need to receive income compensation; this might vary depending on other factors (for example, the existence of inclusive schools). Thus, multidimensional measures provide more information related to the levels of deprivation experienced by individuals. In addition, these measures can provide information regarding the income compensation that a person needs to have a similar level of well-being and which other policies should be implemented to guarantee the inclusion of persons with disabilities in similar conditions to those experienced by a person without a disability. 

## 7. Conclusions

The SoL of persons with disabilities and their families is directly affected by the extra costs associated with disability. However, how much this is reflected in the SoL method depends on the context and the levels of opportunities that individuals in a society have. Although considering using multidimensional poverty measures when analysing the extra costs of disability can be an option, aspects related to the relationship between the extra costs of disability and different levels of deprivation should be explicitly mentioned and discussed. Furthermore, it is important to analyse how income compensation can reduce deprivation and how individual, social, and environmental factors can affect the potential effect of that compensation on the lives of persons with disabilities. Analysing the potential effect of extra costs of disabilities on their levels of well-being will benefit from the use of a multidimensional well-being measure, which is a concept that goes beyond the standard of living and can include aspects related to access to opportunities. This measure aims to capture different aspects of life that are desirable for every individual (with and without disabilities). Given that it is tailored to the context, it is expected that even in countries with lower levels of socioeconomic development, it will be possible to analyse the potential effect of extra costs on the well-being of individuals with disabilities. In addition, this measure goes beyond income and tries to include aspects of life that are affected by direct, indirect, and opportunity costs associated with disabilities and how these limit the opportunities of individuals living with disability in a specific society.

## Figures and Tables

**Figure 1 ijerph-20-02729-f001:**
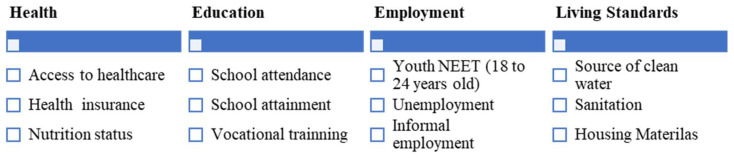
Example of the Multidimensional Poverty Measure. Source: Author’s own elaboration.

**Figure 2 ijerph-20-02729-f002:**
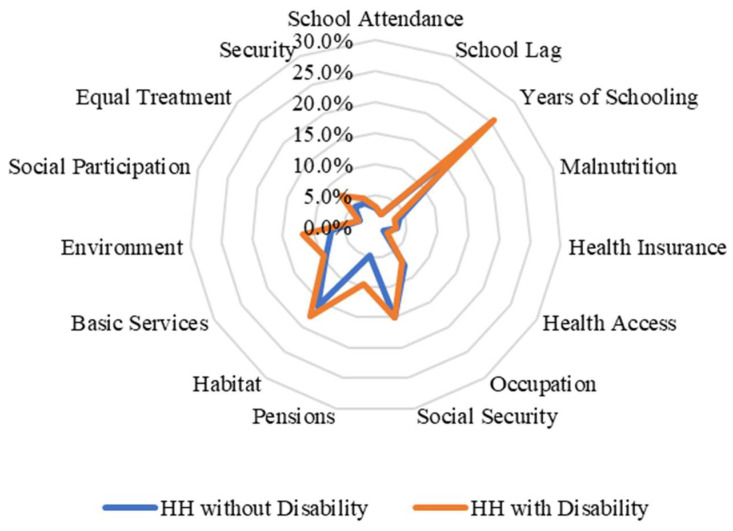
Censored headcount ratios for households with and without members with disability in the lowest quintile in Chile.

**Figure 3 ijerph-20-02729-f003:**
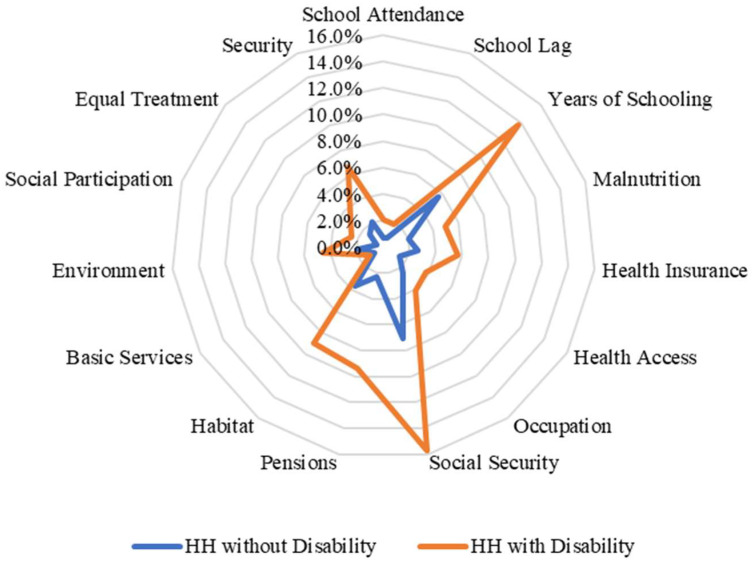
Censored headcount ratios for households with and without members with disability in the highest quintile in Chile.

**Figure 4 ijerph-20-02729-f004:**
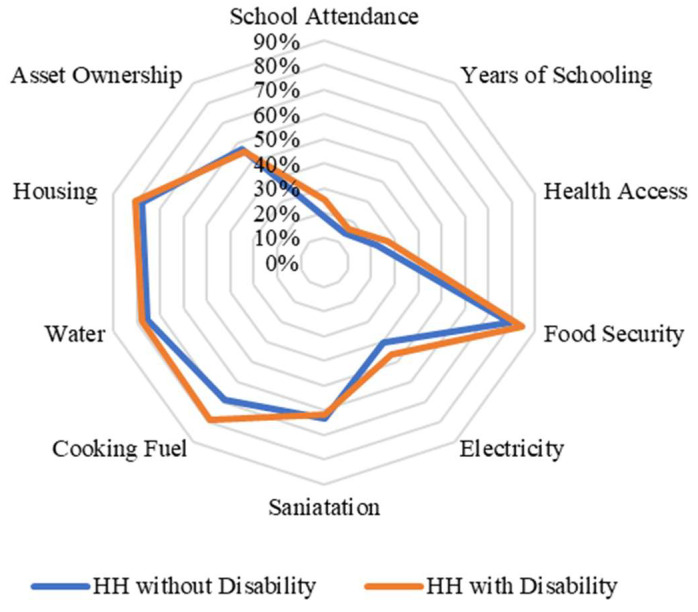
Censored headcount ratios for households with and without members with disability in the lowest quintile in Nigeria.

**Figure 5 ijerph-20-02729-f005:**
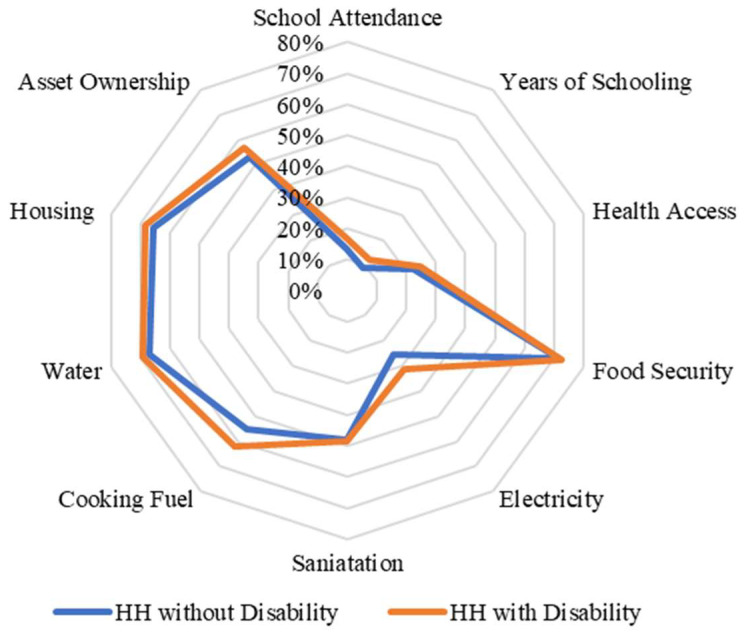
Censored headcount ratios for households with and without members with disability in the highest quintile in Nigeria.

**Table 1 ijerph-20-02729-t001:** Incidence, intensity, MPI, and censored headcount ratios ^a^ by disability status of households in Chile and Nigeria.

Chile			Nigeria		
	HH without Disability	HH with Disability		HH without Disability	HH with Disability
Percentage of…	79.6%	20.4%	Percentage of…	14.5%	85.5%
H (incidence)	18.7%	28.3%	H (incidence)	84.9%	87.7%
A (intensity)	27.7%	28.2%	A (intensity)	51.1%	54.8%
MPI	0.052	0.080	MPI	0.423	0.460
School Attendance	2.0%	3.0%	School Attendance	17.3%	23.6%
School Lag	1.8%	2.4%	Years of Schooling	13.2%	15.5%
Years of Schooling	13.3%	22.0%	Health Access	22.2%	26.6%
Malnutrition	3.2%	4.1%	Food Security	77.7%	81.6%
Health Insurance	3.3%	4.3%	Electricity	37.5%	43.0%
Health Access	1.4%	2.8%	Sanitation	60.4%	58.8%
Occupation	5.3%	6.8%	Cooking Fuel	66.2%	75.4%
Social Security	12.6%	17.6%	Water	74.1%	75.9%
Pensions	3.9%	9.0%	Housing	75.6%	77.8%
Habitat	10.0%	15.6%	Asset Ownership	56.3%	55.5%
Basic Services	4.0%	4.9%			
Environment	4.4%	7.3%			
Social Participation	1.7%	2.9%			
Equal Treatment	3.1%	6.0%			
Security	3.7%	6.7%			

^a^ percentage of individuals who are deprived in an indicator and are multidimensionally poor.

## Data Availability

Data on Nigeria can be found in https://microdata.worldbank.org/index.php/catalog/3827 and data from Chile can be found in http://observatorio.ministeriodesarrollosocial.gob.cl/encuesta-casen-2017.

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
