# Peer review of "Multidimensional Measures and the Extra Costs of Disability: How Are They Related?"

_ijerph, 2023, doi:10.3390/ijerph20032729_

Round 1

Reviewer 1 Report

The aims and the scope of the paper needs to be more clearly and more narrowly defined.

The author states that “This paper aims to analyse if a multidimensional measure can be an option to study the extra costs of disability, and the implications of using a multidimensional measure in this analysis”. It argues that multidimensional poverty cannot be used as a proxy for standard of living in the Standard of Living (SOL) measure of extra costs. I agree with the author that using multidimensional poverty on the left hand side of an SOL model is not appropriate. However, to me this is not what is important about this paper and could be in an appendix to the paper or in a separate paper perhaps as a viewpoint.

More broadly, the paper questions the relationship between multidimensional measures of poverty or wellbeing and extra costs, using the capability approach as a conceptual framework. This broader goal, if achieved, could make this paper very valuable for the literature on the wellbeing or poverty of persons with disabilities. To me this is where the potential contribution of the paper is as the relationship between extra costs and wellbeing/poverty is complex, and yet central to assessments of the situation of persons with disabilities.

Regarding the broader question of the relationship between multidimensional poverty/wellbeing and extra costs, the author importantly notes that “Whether realised or not, extra costs impact the opportunities of individuals with disabilities in general and to access services, that will facilitate income generation and their equal participation in the society.”

In addition, the inability to meet potential extra costs is also central to the paper as it can determine deprivations and thus multidimensional poverty. For instance, one may not be able to attend school due to an inability to meet the costs of assistive devices, so to pay for potential extra costs. This should be stated clearly and explicitly at the start of the paper and then later on when relevant. So far it is only implicit.

Meeting extra costs may also lead to deprivations as well: for instance, spending on transportation may prevent spending on health care.    

Importantly, the authors state that “It is not possible to assume that higher levels of deprivation are only the result of the extra costs that a person might face.” Indeed, in some contexts, extra costs may be completely irrelevant while deprivations are not (i.e. in an environment with very limited income (as shown by the empirical studies in the paper) or unavailable goods/services).

In fact, using the capability approach, the author can deepen this point and argue that the issue of extra costs is secondary to that of wellbeing and poverty. Using the capability approach, what matters ultimately are capabilities and functionings that are valuable in society and to highlight inequalities (Sen 2009) by disability status and other characteristics. One can do that using multidimensional poverty or wellbeing measure. Once that is done, one can figure out if in a given context, extra costs may be correlated with poverty or wellbeing, whether they are causes or consequences of poverty or wellbeing and what policy interventions may be able to cut extra costs and/or deprivations.

I think that a focus on this normative point supported through the capability approach and on an interrogation of the conceptual links between multidimensional poverty or wellbeing would make this paper more focused and more useful to the literature on disability and poverty/wellbeing broadly and on the literature on extra costs, more specifically.

Multidimensional poverty vs wellbeing

An idea in the paper needs to be justified: “A multidimensional wellbeing measure will allow a clearer analysis of the potential loss of welfare than individuals have because of their disabilities.” The authors write later: “In this case, the assumption will not be if individuals with disabilities reduce their consumption of different items, because of the increase in the consumption of disability related items, but instead, the main assumption is that different factors associated with barriers that persons with disability face might reduce their wellbeing. For example, discrimination, stigma, social exclusion, and lack of participation.”

I am not convinced that a wellbeing measure is necessarily better than a multidimensional poverty measure, as one can be thought of as the opposite of the other. The barriers that persons with disabilities face might indeed reduce their wellbeing but at the same time increase their deprivations.

Editing:

The paper needs to be edited throughout. For example:

There is a typo for forthcoming throughout.

Page 1: “

The following sentence and the one that follows it need to be edited for clarifications as it only applies to the GRS method, not the GS method: “In that approach <…> to identify the goods and services they need but might not be able to 32 cover.”

I doubt that “GSR requires expert groups” but it can be done with expert groups. In depth interviews and focus groups of persons with disabilities are also possible methods of data collection.

Page 2:

I think that the Banks et al review covers articles since 2016, not 2017. Please check.

The article has some run on sentences.

Not sure what “transference” means.

Page 4 line 178: remove “Therefore,” in the sentence starting with” “Therefore, monetary measures”

Page 6: on healthcare costs, please consider to cite the new report by WHO on health equity for persons with disabilities which reviews the literature on health outcomes by disability status.

Title” “Indices” in the title is not clear for the reader.

Author Response

The aims and the scope of the paper needs to be more clearly and more narrowly defined.

We appreciate the comments, we have included the aims and scope in page 4, paragraph 2 in the introduction.

The author states that “This paper aims to analyse if a multidimensional measure can be an option to study the extra costs of disability, and the implications of using a multidimensional measure in this analysis”. It argues that multidimensional poverty cannot be used as a proxy for standard of living in the Standard of Living (SOL) measure of extra costs. I agree with the author that using multidimensional poverty on the left hand side of an SOL model is not appropriate. However, to me this is not what is important about this paper and could be in an appendix to the paper or in a separate paper perhaps as a viewpoint.

We appreciate the comment, however, given the current methodologies use to analyse the extra costs of disability, it becomes important to define if using a multidimensional poverty measure can or cannot be used in SoL. Therefore, the article has two main purposes, first to discuss if multidimensional poverty measures can be included in the analysis when using SoL, and second to discuss how multidimensional poverty is related to the extra costs of disability.

In addition, the inability to meet potential extra costs is also central to the paper as it can determine deprivations and thus multidimensional poverty. For instance, one may not be able to attend school due to an inability to meet the costs of assistive devices, so to pay for potential extra costs. This should be stated clearly and explicitly at the start of the paper and then later on when relevant. So far it is only implicit.

We have made explicit this in the introduction and the section on Multidimensional Poverty and its relationship with the extra costs of disability.

I think that a focus on this normative point supported through the capability approach and on an interrogation of the conceptual links between multidimensional poverty or wellbeing would make this paper more focused and more useful to the literature on disability and poverty/wellbeing broadly and on the literature on extra costs, more specifically.

We appreciate the comment, and the article has been restructured aiming to increase the emphasis on how multidimensional poverty relates to the extra costs of disability from conceptual and practical perspectives. Please see the section Multidimensional poverty and its relationship with the extra costs of disability.

Multidimensional poverty vs wellbeing

An idea in the paper needs to be justified: “A multidimensional wellbeing measure will allow a clearer analysis of the potential loss of welfare than individuals have because of their disabilities.” The authors write later: “In this case, the assumption will not be if individuals with disabilities reduce their consumption of different items, because of the increase in the consumption of disability related items, but instead, the main assumption is that different factors associated with barriers that persons with disability face might reduce their wellbeing. For example, discrimination, stigma, social exclusion, and lack of participation.”

I am not convinced that a wellbeing measure is necessarily better than a multidimensional poverty measure, as one can be thought of as the opposite of the other. The barriers that persons with disabilities face might indeed reduce their wellbeing but at the same time increase their deprivations.

We appreciate the comment and we have made the change to emphasis of the paper and highlight the potential use of multidimensional measures in general.

Editing:

The paper needs to be edited throughout.

We appreciate the comment, we have copy-edited the paper.

There is a typo for forthcoming throughout.

Page 1: “

The following sentence and the one that follows it need to be edited for clarifications as it only applies to the GRS method, not the GS method: “In that approach <…> to identify the goods and services they need but might not be able to 32 cover.”

We appreciate the comments and we have corrected the sentence.

I doubt that “GSR requires expert groups” but it can be done with expert groups. In depth interviews and focus groups of persons with disabilities are also possible methods of data collection.

We appreciate the comments and we have corrected the sentence to give more clarity to the idea.

Page 2:

I think that the Banks et al review covers articles since 2016, not 2017. Please check.

We have corrected the dates, Banks et al covers new articles since 2015.

The article has some run on sentences.

We appreciate the comments and we have made the corrections.

Not sure what “transference” means.

We appreciate the comments and we have made the corrections.

Page 4 line 178: remove “Therefore,” in the sentence starting with” “Therefore, monetary measures”

We appreciate the comments and we have made the corrections.

Page 6: on healthcare costs, please consider to cite the new report by WHO on health equity for persons with disabilities which reviews the literature on health outcomes by disability status.

We appreciate the comments and we have included the reference. 

Title” “Indices” in the title is not clear for the reader.

We appreciate the comment we have changed the word for measures.

Reviewer 2 Report

The article addresses the important questions of poverty, inequality and costs of disability. It includes the major debates in the theory and current methodologies and applies the conclusions to comparative datasets for Nigeria and Chile. It is a useful summary article on the topic.

- The rationale for selecting Nigeria and Chile, comparing these countries, and implications for other countries, could be expanded, especially the contextual considerations (personal, social, cultural, policy) impacting the cost of disability.

- The article is long. Some of the background material could be reduced if length is a problem. In its current form the detail is helpful for readers wanting an in-depth summary of the problem, alternatives and implications.

Author Response

Reviewer 2

- The rationale for selecting Nigeria and Chile, comparing these countries, and implications for other countries, could be expanded, especially the contextual considerations (personal, social, cultural, policy) impacting the cost of disability.

We have added more details in page 14 paragraph 1 in the section empirical application.  

- The article is long. Some of the background material could be reduced if length is a problem. In its current form the detail is helpful for readers wanting an in-depth summary of the problem, alternatives and implications.

We appreciate the comment and we have reduced the length of the article, excluding paragraph that were not fundamental to support the main purpose of the article.